# Iodixanol as a New Contrast Agent for Cyanoacrylate Embolization: A Preliminary In Vivo Swine Study

**DOI:** 10.3390/biomedicines11123177

**Published:** 2023-11-29

**Authors:** Kévin Guillen, Pierre-Olivier Comby, Alexandra Oudot, Anne-Virginie Salsac, Nicolas Falvo, Thierry Virely, Olivia Poupardin, Mélanie Guillemin, Olivier Chevallier, Romaric Loffroy

**Affiliations:** 1Department of Vascular and Interventional Radiology, Image-Guided Therapy Center, François-Mitterrand University Hospital, 14 Rue Paul Gaffarel, 21079 Dijon, Francenicolas.falvo@chu-dijon.fr (N.F.); olivier.chevallier@chu-dijon.fr (O.C.); 2ICMUB Laboratory, Bourgogne/Franche-Comté University, 9 Avenue Alain Savary, 21000 Dijon, France; pierre-olivier.comby@chu-dijon.fr; 3Imaging and Preclinical Radiotherapy Platform, Georges-François Leclerc Cancer Center, 1 Rue Professeur Marion, 21079 Dijon, France; aoudot@cgfl.fr (A.O.); mguillemin@cgfl.fr (M.G.); 4Biomechanics and Bioengineering Laboratory, UMR CNRS 7338, Technology University of Compiègne, 60203 Compiègne, France; anne-virginie.salsac@utc.fr; 5Biossan, Pôle Agricole Auxois Sud, 21320 Créancey, France; thierry.virely@biossan.com (T.V.); olivia.poupardin@biossan.com (O.P.)

**Keywords:** embolization, animal, cyanoacrylate, lipiodol, iodinated contrast agent

## Abstract

N-butyl cyanoacrylate (NBCA) is a lipophilic, permanent embolic glue that must be opacified for fluoroscopic guidance. Empirically, lipophilic Lipiodol Ultra Fluid^®^ (LUF) has been added to produce a single-phase physically stable mixture. Varying the dilution ratio allows control of glue polymerization kinetics. LUF is far more costly than water-soluble iodinated contrast agents (ICAs). Our purpose was to evaluate whether a water-soluble nonionic iso-osmolar ICA could be used instead. We embolized both renal arteries of six swine using 1:3 NBCA–LUF or NBCA–iodixanol in 1:1, 1:3, and 1:7 ratios. We used both micro-computed tomography to assess the distality of glue penetration and indexed cast ratio and histology to assess distality, arterial obliteration, vessel-wall damage, and renal-parenchyma necrosis. Glue–LUF produced significantly greater indexed cast ratio and renal-artery ROI values and a significantly shorter cast-to-capsule distance. The injected volume was significantly greater with 1:7 iodixanol than with the other mixtures. No significant differences were found for histological evidence of artery obliteration, vessel-wall damage, or renal-parenchyma necrosis. This is the first study dealing with ICA alone as a contrast agent for cyanoacrylate embolization, compared to LUF. More research is needed to determine whether water-soluble nonionic iodinated agents can be used for human NBCA embolization given the good safety profile, availability, and low cost of ICA.

## 1. Introduction

Cyanoacrylate glues have been used as intravascular embolization agents for over three decades [1,2]. Intravascular embolization is now established as a minimally invasive, effective, and safe alternative to surgery for conditions too numerous to list here. Among embolic agents, cyanoacrylate glues are used in an expanding array of indications [1,2]. Glubran^®^2 (GEM, Viareggio, Italy) is N-butyl cyanoacrylate (NBCA) combined with the monomer metacryloxysulfolane to improve stability and decrease heat release during polymerization upon contact with blood [3]. Glubran^®^2 was the first cyanoacrylate glue to receive the European Conformity mark for endovascular use.

NBCA is radiolucent and is typically diluted with ethiodized oil (Lipiodol Ultra Fluid^®^ [LUF], Guerbet, Aulnay-sous-Bois, France) to enable fluoroscopic guidance. LUF has been reported to provide good visibility during embolization [4]. Moreover, dilution with LUF slows the polymerization rate, and a greater glue dilution, therefore, results in a greater distality of glue penetration along the vascular bed [5,6]. The dilution ratio can thus be chosen according to the specific needs of each patient. The behavior of NBCA–LUF mixtures has been assessed in detail [5]. Both NBCA and LUF are lipophilic compounds and combining them produces a single-phase mixture that can be kept for 30–45 min before use [7]. However, LUF has drawbacks. Less hypersensitivity reactions and renal dysfunction that can be seen with all iodine-containing agents are observed, but LUF has been associated with adverse events, including embolism at various sites [8,9,10,11] and damage to plastic material as 3-way stopcocks and syringes. Moreover, LUF is far more costly than water-soluble iodinated contrast agents (ICAs). 

Other ICAs have, therefore, been evaluated for opacification during cyanoacrylate embolization. Among water-soluble ICAs, iodixanol (Visipaque^®^ 320 mg, GE Healthcare, Chicago, IL, USA) is both iso-osmolar and nonionic, two features associated with decreased toxicity of this ICA [12,13]. Iodixanol is the only iso-osmolar nonionic contrast agent approved for intravascular use [13]. In several randomized trials, iodixanol resulted in less renal toxicity compared to low-osmolar ICAs [14]. We previously reported an in vitro study showing that iodixanol produced water-in-oil emulsions whose micelle size could be modulated by varying the mixing speed [15]. The micelles may act as microparticles, therefore adding to the lumen-obliterating effect of cyanoacrylate. A few studies have evaluated the ICA iopamidol for cyanoacrylate glue opacification, but only in addition to LUF. In two swine studies, at least a 1/6th proportion of iopamidol was required to prevent adhesion within the microcatheter, and the optimal NBCA–LUF–iopamidol ratio was 2/3/1 [16,17,18]. Although iopamidol is a low-osmolar agent, iodixanol is iso-osmolar and might, therefore, be expected to cause a greater delay in NBCA polymerization and, therefore, greater distality of penetration [6,7,19]. To our knowledge, no in vivo study of NBCA mixed with iodixanol for embolization procedures has been published. 

The objective of this study was to compare the results of embolization with Glubran^®^2 glue opacified with LUF vs. iodixanol in several concentrations. We embolized both renal arteries in vivo in swine and then used both micro-computed tomography (µCT) and histology to assess the results.

## 2. Materials and Methods

### 2.1. Swine Renal-Artery Embolization 

Swine closely resemble humans regarding blood flow, renal vessel diameters, and hemostasis. The terminal configuration of the renal arterial vasculature allows an assessment of glue-cast properties, including distribution after injection [6,20]. Therefore, we chose the renal arteries of swine for these embolization experiments.

The experiments were performed at the Biossan Platform (Créancey, France) after approval by the local ethics committee for animal studies (APAFIS ##39508-2022110920223230 V2). The study complied with the guidelines of the European directive on animal experimentation (directive EU/2010/63). We studied six healthy female swine weighing 48–55 kg. In each animal, both renal arteries were embolized for a total of 12 embolization procedures.

A veterinarian (TV) continuously monitored anesthesia depth and physiological parameters throughout the procedures. All experiments were done by two radiologists (RL and KG with 25 years’ cumulative experience) under fluoroscopic guidance (Siemens, Erlangen, Germany). The animals were sedated with intramuscular zolazepam/tiletamine 5 mg/kg (Zoletil 100^®^ Virbac, Carros, France), intubated, and anesthetized using 1% to 1.5% isoflurane (Isoflo^®^, Zoetis, Parsipanny-Troy Hills, NJ, USA). A 6-Fr introducer (Terumo, Leuven, Belgium) was inserted into the common femoral artery under sterile conditions. A baseline abdominal angiogram was obtained by injecting iomeprol (Iomeron^®^ 350 mg iodine/mL, Bracco, Milan, Italy) into a 5-Fr Cobra diagnostic catheter. The two renal arteries were then catheterized successively using two different 2.7-Fr microcatheters (Progreat^®^, Terumo, Leuven, Belgium inserted coaxially through the 5-Fr catheter. Each microcatheter was positioned 1 cm distal to the renal-artery ostium. 

The embolization technique was as follows. At room temperature, the cyanoacrylate glue Glubran^®^2 was mixed manually with either LUF or iodixanol using an LUF-resistant three-way stopcock (Vectorio^®^, Guerbet, Aulnay-sous-Bois, France). The NBCA:LUF ratio was 1:3, whereas iodixanol was tested in ratios of 1:1, 1:3, and 1:7. Three procedures were done with each of these four mixtures. With iodixanol, notably at the highest concentration (7 parts to 1 part glue), the mixture was difficult to inject, as it became rubbery in the syringe (Figure 1). 

The microcatheter was flushed with 5% dextrose to remove any anions potentially responsible for premature glue polymerization. The glue–ICA mixture was injected manually under free-flow conditions using a 5-mL syringe (Plastipak^®^, Becton Dickinson Plastic, Franklin Lakes, NJ, USA). The injected volume was recorded. The end point was considered to be glue backflow on the microcatheter distal tip. This was monitored by fluoroscopy. The device was then immediately retracted to any adhesion to the vessel wall. A new microcatheter was needed for the contralateral procedure. At the end of the procedure, the vascular access site was closed, and the animal was euthanized using an intravenous bolus of 0.5 mL/kg of sodium pentobarbital (Euthasol^®^, Le Vet Pharma, Oudewater, The Netherlands) while still under general anesthesia. The kidneys were harvested and fixed in 4% formaldehyde (Figure 2). The intravascular distribution of the embolic mixture in the kidneys was assessed 24 h later by µCT. Pathological studies of kidney samples were then performed. 

### 2.2. Micro-Computed Tomography

#### 2.2.1. Acquisition and Reconstruction Parameters 

We used the µCT component of the NanoSPECT/CT Plus™ small-animal camera (Bioscan, Poway, CA, USA). A scouting scan was first obtained to define the axial examination range. Then, the µCT was acquired with the following parameters: 55 kV, 1000 ms exposure time for each of the 240 projections, 143–179 mm axial range, and pitch of 1. The acquisition time was 14–16 min.

The µCT reconstructions were performed using Bioscan image-processing software. An exact cone-beam Filtered Back Projection (FBP) algorithm produced reconstructed image slices with a voxel size of 74 × 74 × 147 µm^3^. A Ramlak filter was applied with the cutoff frequency set at the Nyquist frequency. Each reconstructed slice was opened using ViVoquant™ software (Invicro, Needham Heights, MA, USA) and carefully oriented before each measurement. All measurements were made by the same investigator (KG).

#### 2.2.2. Objectively Assessed µCT Outcome Measures

The embolized kidney areas were segmented, starting at the first renal-artery bifurcation to avoid cast-volume variability linked to trunk length, as previously described [6]. In the first segment, the cast was sub-segmented according to whether attenuation was 1000 to 1300 HU, >1300 to 1600 HU, or >1600 HU. These cutoffs were chosen based on the high attenuations of LUF and other ICAs, which create a risk of streak artifacts. We validated this one in a past work for Lipiodol and cyanoacrylate mixture [6]. We assumed that all pixels whose attenuation was greater than that of the renal parenchyma were from embolized vessels. To limit variability due to differences in individual characteristics of the animals (e.g., weight and age), we computed the indexed cast ratio by normalizing each segmented cast volume for the corresponding renal-parenchyma segment volume. 

Cast density in the renal-artery trunk (HU) was also evaluated and named cast artery ROI. Post-embolization, artery diameter and cast-to-capsule distance (mm) were measured in the middle and at the upper and lower poles of the kidney to assess the depth of glue–ICA penetration further.

#### 2.2.3. Subjectively Assessed µCT Outcome Measures

Cast heterogeneity was scored semi-quantitatively in the first three distribution segments, as follows: 0, no opac cast in the vessel lumen; 1, opac cast in ≤25% of the lumen, 2, opac cast in >25% to ≤50%; 3, opac cast in >50% to ≤75%; and 4, opac cast in >75%. Given uncertainties regarding the assessment of distality linked to the lack of special resolution, only the heterogeneity scores in the first two distribution segments were considered in the statistical analysis. 

We also assessed the presence of contrast in the renal calyces as present or absent. 

### 2.3. Histological Evaluation

The kidneys and renal arteries were collected after euthanasia and fixed in 10% formol for 48 h. The samples were handled by the Oncovet Clinical Research Corporation (Loos, France). After trimming, each kidney was cut in half in the coronal plane, and the ventral half was cut into four parts (cranial, caudal, cranio–ventral, and caudo–ventral). The ventral half-kidney was sectioned into four parts (cranial, caudal, cranio–ventral, and caudo–ventral) to obtain samples representative of all the renal-artery branches. The main renal artery was sectioned transversally. The kidney and renal-artery samples were then embedded in paraffin (Tissue-Tek^®^ Paraffin, Sakura, Torrance, CA, USA), cut into 5 µm-thick slices, and stained with hemalum-eosin (HE) (Prisma, Sakura). This process produced 70 slides, which were evaluated under light microscopy by an experienced veterinary pathologist blinded to the glue–ICA used. 

The pathologist recorded the distribution and appearance of the injected material for each slice. A semi-quantitative score was used to describe the distality of glue–ICA distribution along the arterial tree: 0, no visible cast; 1, renal artery and first branches; 2, interlobar artery; 3, corticomedullar junction; 4, interlobular arteries in the deep cortex; and 5, interlobular arteries in the superficial cortex (Figure 2). Artery obliteration by the glue–ICA was scored semi-quantitatively as follows: 0, fully patent artery; 1, partial obliteration; 2, nearly complete obliteration; and 3, complete obliteration. To assess vessel-wall damage, the pathologist applied the Banff classification [21] originally designed for renal allograft grading: 0, no changes; 1, intimal arteritis; 2, alterations of the media; and 3, transmural arteritis. Finally, the presence of renal-parenchyma necrosis and/or degeneration was also scored semi-quantitatively, given the ill-defined borders between abnormal and normal tissue. The scale was as follows: 0, no lesions; 1, limited focal lesions; 2, cohesive multifocal lesions; and 3, extensive lesions. 

### 2.4. Statistical Analysis

For continuous variables, we computed mean ± SD. To avoid overfitting, we analyzed only a cast attenuation ≥1300 HU because all cast ratios were interdependent. Semi-quantitative variables were described as median [interquartile range]. Semi-quantitative data (e.g., cast heterogeneity and pathology scores) were handled as ordinal variables. 

We first assessed the dose-response effect using the single glue–LUF mixture (1:3) as the reference. We then performed binary comparisons of the glue–iodixanol groups (1:1, 1:3, and 1:7) vs. the glue–LUF group for each study variable. Ordinal regression was used for ordinal variables. Logistic regression was performed for dichotomous variables, and linear regression was applied for quantitative variables. Significant *P* values were defined by a cutoff of 0.05 or less. The statistical analyses were performed using STATA software version 15.1 (STATA, College Station, TX, USA).

## 3. Results

### 3.1. Micro-Computed Tomography Parameters

#### 3.1.1. Objectively Assessed µCT Parameters

Table 1 reports the results for the objectively assessed parameters.

The ratio was higher with glue–LUF than with glue–iodixanol, whereas values were similar among the three iodixanol mixtures. The wide variability in cast-to-capsule distance, used to assess the distality of glue–ICA penetration, was largely due to the lesser penetration of the 1:1 glue–iodixanol mixture compared to the other three mixtures (mean cast-to-capsule distances, 11.7 mm vs. 3.9 mm). Post-embolization renal-artery diameter showed little difference across the four mixtures. The injected volumes ranged from 1.8 to 2.8 mL after the exclusion of the 1:7 glue–iodixanol mixture, for which the volumes were considerably higher (7, 7, and 4.8 mL, respectively). Figure 3 shows examples of µCT images of kidneys embolized with three different mixtures, and Figure 4 shows a kidney embolized with glue–iodixanol in a 1:7 ratio.

Left panel: Kidney embolized with glue–iodixanol in a 1:7 ratio; note the opacification of the cortex and the arrow pointing to the opacified renal pelvis. Middle panel: Kidney embolized with glue–LUF (1:3 ratio); note the better opacification of the cast in the vascular tree. Right panel: Kidney embolized with glue–iodixanol in a 1:1 ratio; note the heterogeneous appearance of the cast in the renal artery.

#### 3.1.2. Subjectively Assessed µCT Parameters

Cast heterogeneity displayed a high variability with a median evaluated at 1.5 with an IQR that goes from 0.5 to 3.

### 3.2. Histological Parameters

The quality of the histological preparations was satisfactory overall, as shown in Figure 5. Cracks or folds were present in some cases and resulted in the exclusion of eight samples. 

The glue–ICA mixture consistently reached the interlobular arteries of the superficial cortex (maximum score of 5). Glue–ICA was present in the calyces in all cases. Figure 6 shows the appearance of the glue–ICA in the arterial tree.

In some cases, the glue–ICA was visible only after the removal of artifacts (Figure 6, top panel). The embolized segments appeared as regular empty spaces in the arterial lumen. Iodixanol micelles were visible, notably with the 1:3 ratio (Figure 6, bottom panel).

Figure 7 shows the results of the semi-quantitative assessments of artery obliteration, vessel-wall damage, and renal-parenchyma necrosis. Artery obliteration was nearly complete (score of 2) or complete (score of 3) in 78% of kidney slices. All three levels of vessel-wall damage were observed (intimal endothelial-cell degeneration, fibrinoid necrosis or inflammatory-cell infiltration of the media, and transmural arteritis (Figure 6). Periarterial edema or bleeding indicating permeability of the damaged vessel walls was seen in some cases. The Banff scores showed that vessel-wall damage chiefly involved the smaller downstream arteries and that most renal arteries exhibited little or no damage: the Banff score was 0 or 1 for 40% of the small intra-parenchymal arteries versus 94% of the larger renal arteries.

The renal parenchyma in embolized areas exhibited multifocal necrosis of the tubular parietal cells, chiefly in the corticomedullary and medullary zones. These lesions varied widely across slices in both intensity and extent. The score was 0 (no lesions) in 38% of slices. In some cases, necrotic-cell debris was seen within dilated tubules. Focal inflammatory-cell infiltrates were occasionally visible near necrotic areas but were possibly related to pre-existing sub-acute interstitial nephritis, a common nonspecific finding in swine.

### 3.3. Dose-Response Effect and Binary Analysis

Table 2 reports the results of the dose-response analysis with LUF 1:3 glue–LUF considered to be a reference compared to the impact of iodixanol dilution. Histological obliteration of the arteries was not significantly greater with glue–LUF than with 1:7 glue–iodixanol. Cast-to-capsule distance was significantly shorter, suggesting a greater distality of penetration, with 1:3 glue–LUF. No other significant associations with cyanoacrylate concentration were found. 

Table 3 shows the results of the binary comparisons. The indexed cast ratio and renal-artery ROI were significantly greater with glue–LUF than with the iodixanol mixtures, probably due to the greater radio opacity of LUF. A significant difference was also found for cast-to-capsule distance.

## 4. Discussion

In this in vivo swine study of renal embolization using the cyanoacrylate glue Glubran^®^2, the histological score for arterial obliteration was not significantly different between the mixture containing one quarter glue and three quarters LUF and the mixture containing several ratios of glue and iodixanol. However, distality appeared greater with the 1:3 glue–LUF mixture, as shown by the shorter cast-to-capsule distance. The 1:7 glue–iodixanol emulsion was associated with significantly greater injected volumes. In the binary analysis, compared to the iodixanol mixtures, the glue–LUF produced significantly greater indexed cast ratio and renal-artery ROI values and a significantly shorter cast-to-capsule distance. Nonetheless, no differences were found between mixtures considering arteritis or necrosis histological scales. 

To our knowledge, this is the first study of a water-soluble iodinated compound used as the only contrast agent during cyanoacrylate embolization. Other studies have assessed the addition of the low-osmolar water-soluble nonionic ICA iopamidol to glue plus LUF [16,17,18]. Consequently, there is no earlier data to which ours can be compared.

We used two complementary assessment tools, µCT and histology. Several studies have used µCT to assess the intravascular [11,22] or extravascular [23] distribution of compounds. By adding histology, we were able both to confirm the distribution pattern demonstrated by µCT and to obtain evidence of embolization effectiveness in stopping the arterial blood supply.

In our µCT assessment, we evaluated the distality of glue penetration based only on the cast-to-capsule distance. This method was able to detect significant differences in distality across doses and between LUF and iodixanol. Moreover, the injected volume was greater, with the lowest glue concentration (1:7 glue:iodixanol). These findings are consistent with earlier data obtained using µCT [6] or histology [5,24,25]. 

We computed the indexed cast ratio to limit variability due to animal age and weight. Also, this parameter avoided overestimation of cast volume by the inclusion of calyces opacified during the pre-embolization angiography. The absence of a significant dose-effect response for the indexed cast ratio may be ascribable to the limited statistical power produced by the small number of procedures. The indexed cast ratio was significantly greater with glue–LUF than with the iodixanol mixtures, probably because of the greater radio opacity of LUF. The cutoff needs to be adjusted for iodixanol because, in iodixanol, there is one third less iodine per milliliter compared to LUF. By histology, the effects of embolization were similar to all the mixtures tested. 

Vascular tree dilation was limited in our study, in contrast to our previous findings in a rabbit model of renal-artery embolization [6]. However, in the rabbit model, half the procedures were done using blocked-flow injection, whereas the present study used only free-flow injection, which would be expected to induce less dilation. The more limited vessel-wall damage and renal-parenchyma lesions compared to the rabbit-model study are also consistent with the use of free-flow only. Moreover, the foci of necrosis were due to the glue–ICA injection and not to ischemia induced by mechanical compression. Another factor that may have decreased vessel-wall damage and renal-parenchyma necrosis is that euthanasia was performed immediately after the second embolization procedure, compared to 30 min later in the rabbit model. Interestingly, although swine weigh 10 to 20 times more than rabbits, the injected volume in swine was only 2- to 3-fold that in rabbits [5,6].

Our histological study suggested the presence of micelles with iodixanol. This ICA forms a water-in-oil emulsion in cyanoacrylate at a 1:3 glue–iodixanol ratio [15]. Conceivably, these micelles may contribute to occlude the blood vessels as microparticles. However, compared to the single-phase glue–LUF mixture, the glue–iodixanol emulsion was less stable and acquired a paste-like consistency over time with the highest iodixanol concentrations after about 8 to 10 min. This behavior may be a limitation in clinical practice. Nonetheless, considering only ICA dilution, the cast-to-capsule distance was shortest with 1:7 glue–iodixanol, probably linked to the considerably larger volume injected. Whether phase inversion can occur with glue-water-soluble ICA emulsions requires investigation. 

LUF is well known to exert toxic effects. Among water-soluble nonionic ICAs, those in the low-osmolar and iso-osmolar categories have been reported to induce less renal toxicity compared to their high-osmolar counterparts when used alone. In randomized head-to-head comparisons, the iso-osmolar ICA iodixanol had a better renal safety profile than did low-osmolar ICAs [14]. A meta-analysis, however, showed similar renal toxicity of iodixanol and several low-osmolar ICAs [25,26]. Moreover, an in vitro study suggested greater changes in blood-component configuration with iodixanol than with low-osmolar ICAs [14]. These changes might increase the risk of heart dysfunction and thrombosis. In a meta-analysis, when only the randomized trials (*n* = 11) were considered, in-hospital cardiovascular events were less common with iodixanol than with low-osmolar agents. Of note, the glue–iodixanol mixture being a water-in-oil emulsion, the iodixanol micelles are coated by glue and, therefore, not in direct contact with blood [15,27].

Our work has limitations. First, only three procedures were done with each mixture, resulting in low statistical power that might have prevented the detection of significant differences, notably for the indexed cast ratio. However, restricting the number of animals used in experimental studies is one of the 3Rs ethical principles of animal welfare [28]. Our decision to compare LUF to a single water-soluble ICA was also taken to decrease the number of animals needed. Second, the animals were euthanized immediately after embolization of the second renal artery, which may not have allowed sufficient time for renal-parenchyma lesions to develop. Nonetheless, the time to euthanasia was the same for all four mixtures and, therefore, could not have altered the comparative results. Moreover, performing euthanasia while the animal is still anesthetized is likely to minimize suffering. Third, we performed all the embolization procedures using the free-flow technique. The effects of blocked-flow injection with iodixanol will need to be investigated.

## 5. Conclusions

Iodixanol alone, used to opacify cyanoacrylate glue during transarterial embolization, was similarly effective to LUF regarding arterial obliteration and distality of penetration, as assessed by histology. Moreover, µCT displayed a dose-effect response on distality and cast volume. Given the good safety profile and low cost of iodixanol, these findings are very promising and of utmost importance. However, further studies are needed to characterize the behavior of cyanoacrylate glue–iodixanol mixtures in vivo and to determine the optimal dilution ratio.

## Figures and Tables

**Figure 1 biomedicines-11-03177-f001:**
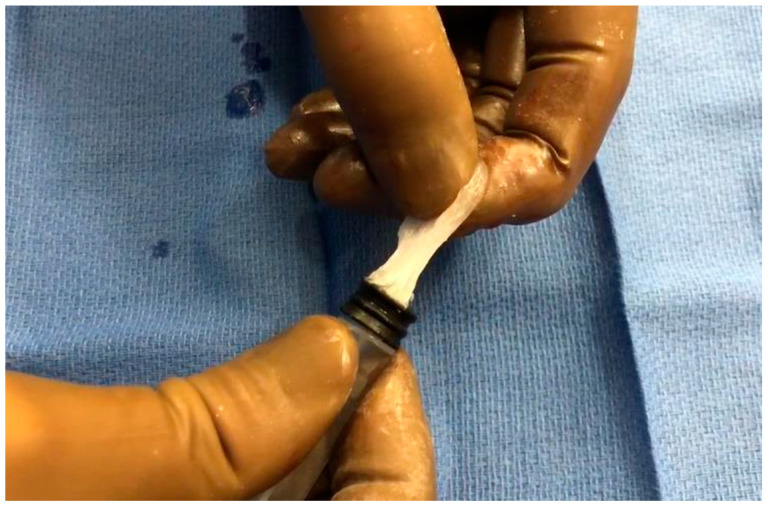
Rubbery consistency of the glue–iodixanol mixture, notably with the 1:7 ratio.

**Figure 2 biomedicines-11-03177-f002:**
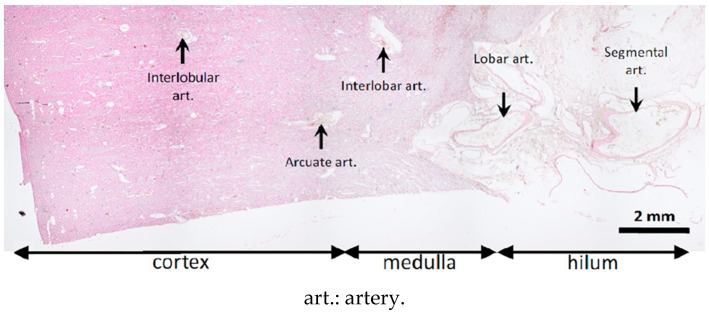
Sites assessed for glue–ICA penetration on histological slices.

**Figure 3 biomedicines-11-03177-f003:**
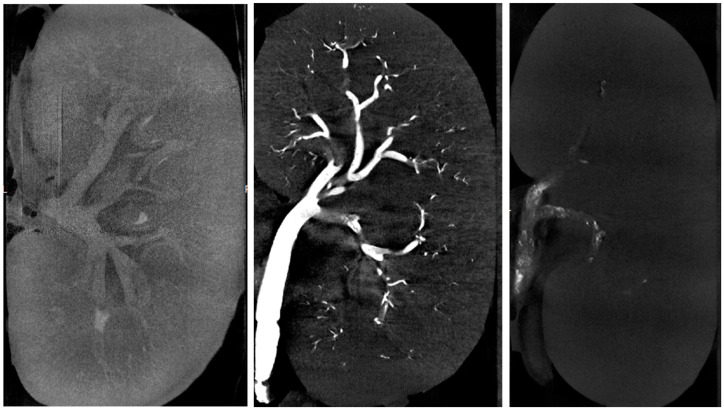
Micro-computed tomography of an embolized swine kidney.

**Figure 4 biomedicines-11-03177-f004:**
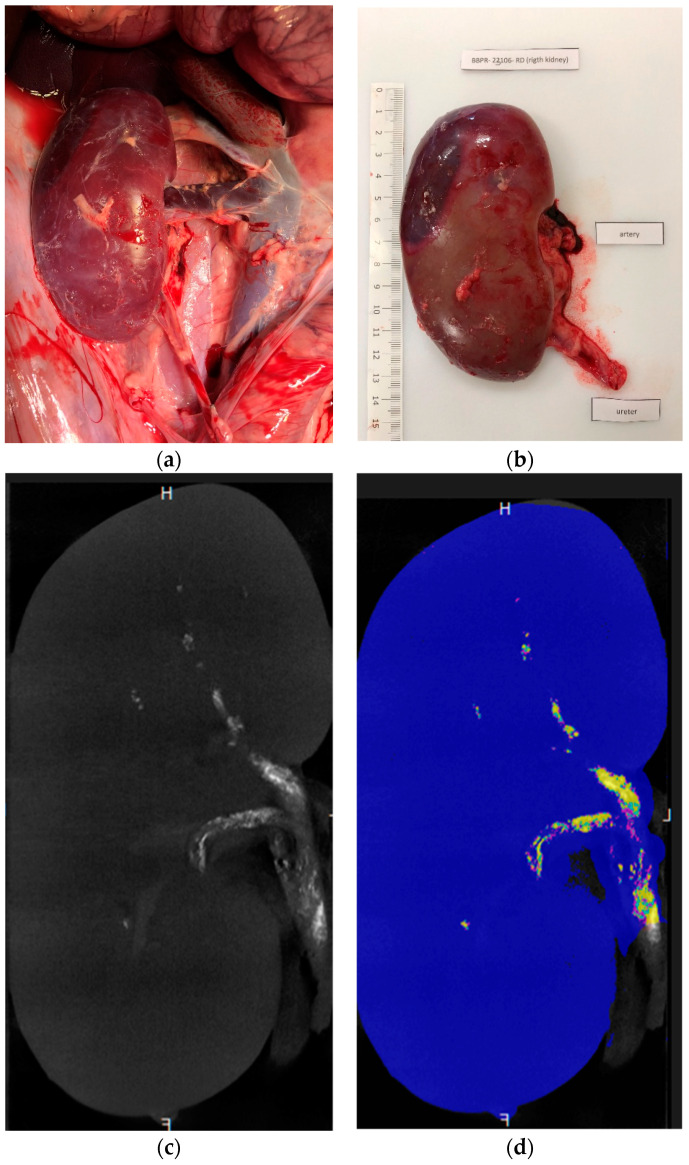
Kidney embolized with 1:7 glue–iodixanol. (**a**) In situ laparoscopic view. (**b**) Appearance after excision. (**c**,**d**) Coronal ex vivo µCT images (**c**) before maximum intensity projection reconstruction, used for a more detailed assessment of fragmentation, and (**d**) post-processing segmentation (blue, renal-parenchyma segmentation; purple, cast segmentation from 1000 to 1300 HU; cyan blue, cast segmentation from 1300 to 1600 HU; and yellow, cast segmentation above 1600 HU).

**Figure 5 biomedicines-11-03177-f005:**
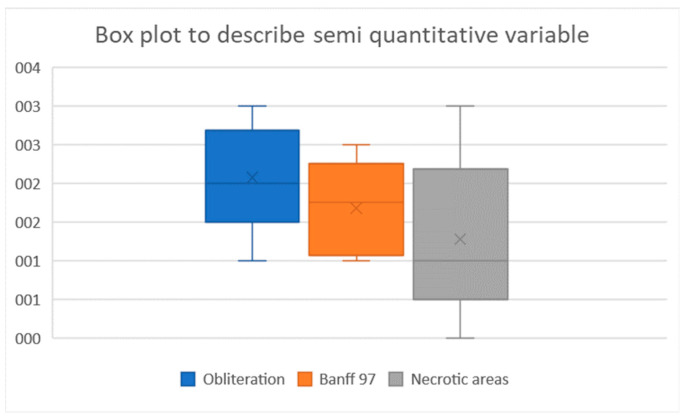
Box plots of the semi-quantitative parameters.

**Figure 6 biomedicines-11-03177-f006:**
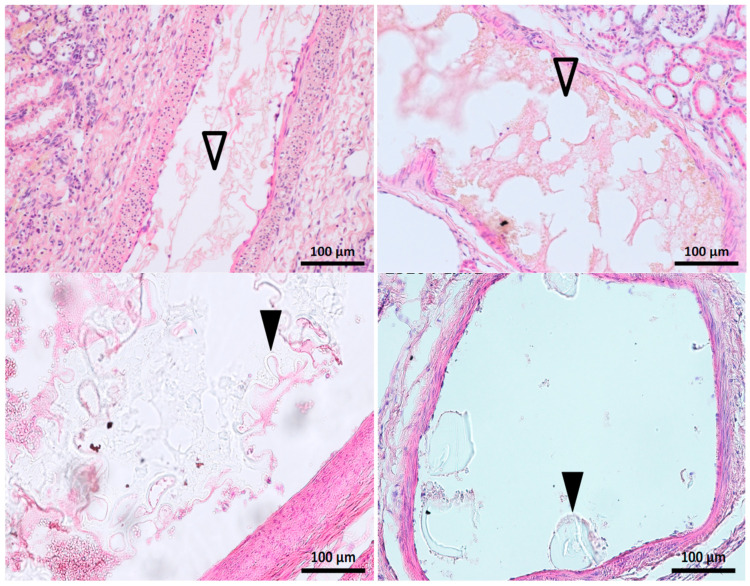
Histological appearance of the glue–ICA mixture in the renal arterial tree. Top panel: Longitudinal slice, 1:3 glue–LUF on the left and 1:7 glue–iodixanol on the right. The embolized segments are seen as evenly contoured, empty spaces in the arterial lumen (Δ) mixed with fibrin and blood cells; micelles are visible on the right, with the iodixanol. Bottom panel: 1:3 glue–iodixanol. Remnants of the embolic mixture are visible in the arterial lumen as meshes or plugs (▲) of refringent material faintly stained by hemalum-eosin.

**Figure 7 biomedicines-11-03177-f007:**
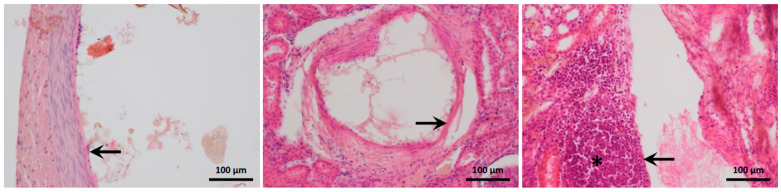
Histological appearance of the arterial walls. The occluded arteries are dilated. Intimal endothelial-cell degeneration is visible in the left-hand panel (1:1 glue–iodixanol, arrow), fibrinoid necrosis in the middle panel (1:3 glue–iodixanol, arrow), and transmural arteritis in the right-hand panel (1:3 glue–LUF, arrow). The asterisk indicates perivascular mixed inflammatory-cell infiltration. Hemalum-eosin stain.

**Table 1 biomedicines-11-03177-t001:** Objectively assessed µCT parameters.

Parameters	Mean ± SD (Range)
Indexed cast ratio, %	1.10 ± 1.31 (0.07–3.36)
Renal-artery ROI (HU)	2955.33 ± 2456.21 (204.16–6918.28)
Cast-capsule distance, mm	5.84 ± 460 (0.83–1467)
Post-embolization renal-artery diameter, mm	461 ± 1.10 (2.80–680)
Volume of glue–ICA injected, mL	3.30 ± 1.92 (1.80–7.00)

ROI: region of interest; ICA: iodinated contract agent (Lipiodol Ultra Fluid or iodixanol).

**Table 2 biomedicines-11-03177-t002:** Dose-response effect.

	SE	*p* > |t|	95% CI	
Obliteration	0.86	0.064	−4.09	0.15
Banff score	0.55	0.456	−0.95	1.85
Necrosis	0.89	0.347	−1.33	3.16
Indexed cast ratio	0.02	0.095	−0.08	0.01
Renal-artery ROI	4115.78	0.378	−13,902.78	6090.25
Cast heterogeneity	2.42	0.880	−5.51	6.27
Cast-capsule distance	2.67	0.001	10.78	24.33
Post-embolization renal-artery diameter	1.82	0.454	−2.99	5.91
Glue–ICA volume injected	3.04	0.206	−11.68	3.09

SE: standard error; 95%CI: 95% confidence interval; ROI: region of interest; ICA: iodinated contrast agent.

**Table 3 biomedicines-11-03177-t003:** Binary comparisons.

	SE	*p* > |t|	95%CI	
Obliteration	0.48	0.217	−1.72	0.45
Banff score	0.25	0.340	−0.35	0.88
Necrosis	0.43	0.277	−0.54	1.57
Indexed cast ratio	0.00	0.000	−0.04	−0.02
Renal-artery ROI	974.49	0.001	−6956.25	−2567.02
Cast heterogeneity	1.07	0.376	−1.42	3.42
Cast-capsule distance	2.50	0.042	0.28	12.16
Post-embolization renal-artery diameter	0.83	0.365	−2.69	1.10
Glue–ICA volume injected	1.55	0.638	−2.75	4.26

SE: standard error; 95%CI: 95% confidence interval; ROI: region of interest; ICA: iodinated contrast agent.

## Data Availability

Additional data are available upon request.

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
