# Peer review of "Iodixanol as a New Contrast Agent for Cyanoacrylate Embolization: A Preliminary In Vivo Swine Study"

_biomedicines, 2023, doi:10.3390/biomedicines11123177_

Round 1
Reviewer 1 Report
Comments and Suggestions for Authors
The authors provided a comprehensive proposal of Iodixanol as a new contrast agent for cyanoacrylate embolization. The methods are well exposed and detailed. Measurements, histological evaluation, and statistical analysis are clearly detailed.
As such it is a very comprehensive study. The discussion clarifies the usefulness of the findings, highlighting advantages and practical points of this contrast agent.
Comments on the Quality of English LanguageEnglish is appropriate and formally correct.
Author Response
Response to Reviewer 1 comments
The authors provided a comprehensive proposal of Iodixanol as a new contrast agent for cyanoacrylate embolization. The methods are well exposed and detailed. Measurements, histological evaluation, and statistical analysis are clearly detailed.
As such it is a very comprehensive study. The discussion clarifies the usefulness of the findings, highlighting advantages and practical points of this contrast agent.
Reply : Thank you very much for your comment. Nothing to add.
Comments on the Quality of English Language
English is appropriate and formally correct.
Reply : Thank you very much for your comment. Nothing to add.
Reviewer 2 Report
Comments and Suggestions for Authors
Iodixanol as a New Contrast Agent for Cyanoacrylate Embolization: An In Vivo Swine Study
The manuscript topic is important and socially significant, supported with experiment. The work is valuable for its practical focus. This is experimental work, with very useful conclussions.
1. In my opinion the title must be changed - Iodixanol as a New Contrast Agent for Cyanoacrylate Embolization: A Preliminary In Vivo Swine Study
The reason for this is a small number of tested animals – only six. This leads to the lack of scientific depth.
2. The Abstract and Conclusions must be improved.
Improvement. In the abstract and conclusions must be underline clearly the new results and conclusions presented from the authors which differ from those obtained till now – add the sentence in row 343-344 - “To our knowledge, this is the first study of a water-soluble iodinated compound used 343 as the only contrast agent during cyanoacrylate embolization”. In the manuscript text there are these first time benefits. In the Abstract must be included the main conclusions and the authors must underline the own approach contributions and the basic benefits from presented results in practice. The unwritten rule is that most readers only look at these paragraphs – abstract and conclusions.
3. Row 40. CE – missed definition?
I hope that the proposed corrections will increase the quality of the manuscript and possibly its citability.
Author Response
Response to Reviewer 2 comments
Comments and Suggestions for Authors
Iodixanol as a New Contrast Agent for Cyanoacrylate Embolization: An In Vivo Swine Study
The manuscript topic is important and socially significant, supported with experiment. The work is valuable for its practical focus. This is experimental work, with very useful conclusions.
Reply : Thank you very much for your comment. Nothing to add.
- In my opinion the title must be changed - Iodixanol as a New Contrast Agent for Cyanoacrylate Embolization: A PreliminaryIn Vivo Swine Study
The reason for this is a small number of tested animals – only six. This leads to the lack of scientific depth.
Reply : Thank you very much for your comment. We fully agree. It has been changed as suggested.
- The Abstract and Conclusions must be improved.
Improvement. In the abstract and conclusions must be underlined clearly the new results and conclusions presented from the authors which differ from those obtained till now – add the sentence in row 343-344 - “To our knowledge, this is the first study of a water-soluble iodinated compound used 343 as the only contrast agent during cyanoacrylate embolization”. In the manuscript text there are these first time benefits. In the Abstract must be included the main conclusions and the authors must underline the own approach contributions and the basic benefits from presented results in practice. The unwritten rule is that most readers only look at these paragraphs – abstract and conclusions.
Reply : Thank you very much for your comment. All changes have been added as suggested in the abstract and the conclusion sections.
- Row 40. CE – missed definition?
Reply : Thank you very much for your comment. It has been corrected. This is European Conformity (EC).
I hope that the proposed corrections will increase the quality of the manuscript and possibly its citability.
Reply : Thank you very much for your comment. We hope the same.